# Cyclic AMP signalling and glucose metabolism mediate pH taxis by African trypanosomes

Sebastian Shaw [1,2], Sebastian Knüsel [1,3], Daniel Abbühl[1,3], Arunasalam Naguleswaran [1,3], Ruth Etzensperger[1], Mattias Benninger[1] & Isabel Roditi [1✉]

The collective movement of African trypanosomes on semi-solid surfaces, known as social motility, is presumed to be due to migration factors and repellents released by the parasites. Here we show that procyclic (insect midgut) forms acidify their environment as a consequence of glucose metabolism, generating pH gradients by diffusion. Early and late procyclic forms exhibit self-organising properties on agarose plates. While early procyclic forms are repelled by acid and migrate outwards, late procyclic forms remain at the inoculation site. Furthermore, trypanosomes respond to exogenously formed pH gradients, with both early and late procyclic forms being attracted to alkali. pH taxis is mediated by multiple cyclic AMP effectors: deletion of one copy of adenylate cyclase ACP5, or both copies of the cyclic AMP response protein CARP3, abrogates the response to acid, while deletion of phosphodiesterase PDEB1 completely abolishes pH taxis. The ability to sense pH is biologically relevant as trypanosomes experience large changes as they migrate through their tsetse host. Supporting this, a CARP3 null mutant is severely compromised in its ability to establish infections in flies. Based on these findings, we propose that the expanded family of adenylate cyclases in trypanosomes might govern other chemotactic responses in their two hosts.

[1] Institute of Cell Biology, University of Bern, Bern, Switzerland. [2] Graduate School of Cellular and Biomedical Sciences, University of Bern, Bern, Switzerland. [3] These authors contributed equally: Sebastian Knüsel, Daniel Abbühl, Arunasalam Naguleswaran. ✉email: isabel.roditi@unibe.ch

Environmental sensing, chemotaxis, and quorum sensing present possibilities for pathogens to orient themselves within their hosts. All three are widespread in bacteria, and the sensors and response mechanisms have been characterised in detail[1,2]. Genes involved in bacterial chemotaxis and quorum sensing have long been linked to biofilm formation and swarming behaviour[3,4]. By contrast, very little is known about chemotaxis in unicellular eukaryotes, with the exception of *Dictyostelium discoideum*, which generates and reacts to cyclic AMP (cAMP) gradients, promoting aggregation and differentiation[5–8].

Intracellular pathogens frequently exploit receptor-ligand interactions to recognise and invade the appropriate cell types[9–12]. For extracellular parasites such as *Trypanosoma brucei spp.*, causative agents of human and animal trypanosomiasis, it is not known what drives them to move from one host tissue to another. In their mammalian hosts, quorum sensing regulates the titre of *T. brucei* by triggering the differentiation of proliferative slender forms to quiescent stumpy forms[13,14]. When these are taken up by tsetse flies during a blood meal[15] they differentiate to early procyclic (midgut) forms and then to late procyclic forms[16]. In addition to the surface marker GPEET procyclin, which is only expressed by early procyclic forms, several metabolic enzymes, nutrient transporters and signal transducers are differentially expressed between the two forms[17,18]. Colonisation of the midgut by procyclic forms is followed by additional rounds of differentiation as parasites invade the proventriculus, and then move to the salivary glands before being transmitted to a new mammalian host. At present, however, the cues allowing trypanosomes to orient themselves within their hosts are unknown.

Early procyclic forms engage in a form of group migration, known as social motility (SoMo), when cultured on semi-solid surfaces such as agarose plates[18,19]. The parasites initially remain at the inoculation site, dividing approximately once every 24 h, then the communities start to form regularly spaced projections that extend by ~1 cm (500 body lengths) per day. Communities on the same plate are able to sense each other and reorient to avoid contact (Supplementary Movie 1). Late procyclic forms do not exhibit SoMo, but they do produce a repellent that is sensed by early procyclic forms[18]. Communities on plates also react to other signals. Trypanosomes depleted of the spliced leader core protein SmD1 shed exosomes that act as repellents[20], while bacterial colonies can act as attractants[21]. Various species of *Leishmania*, parasites related to trypanosomes, respond chemotactically to a range of compounds including amino acids, carbohydrates and sera, in liquid culture[22–25]. In none of these cases is it known which genes mediate the responses.

Few genes have been implicated in SoMo. Concomitant depletion of adenylate cyclases ACP1&2, or depletion of ACP6 on its own, resulted in a hyper-social phenotype (hyper-SoMo) where the communities formed more projections and migrated more rapidly, suggesting that these are negative regulators[26]. Interestingly, ACP1 and ACP6 are upregulated in late procyclic forms[17], which are SoMo-negative[18]. The flagellar phosphodiesterase B1 (PDEB1)[19] and the endoplasmic reticulum protein Rft1[27,28] are positive regulators of SoMo. Importantly, null mutants of PDEB1 and Rft1 were not only impaired in SoMo, but also defective in colonising the tsetse midgut[27,29]. We postulated that SoMo is largely a chemotactic response to factors produced by the trypanosomes themselves. Here we show that procyclic forms can generate pH gradients through glucose metabolism and that the response of early and late procyclic forms to these gradients can explain the self-organising properties of communities on plates. In addition, we identified two cAMP signalling components involved in the response to acid, ACP5 and the cAMP response protein CARP3. Notably, pH sensing is completely abolished in PDEB1 knockout, where cAMP levels are perturbed.

pH taxis could be biologically relevant as there is a pH gradient in the tissues that are colonised by the parasites during their transmission through the insect host[30]. In support of this, we show that CARP3 null mutant parasites are severely compromised in their ability to establish fly infections.

## Results

**Parasite-secreted factors change properties of the medium and repel migrating projections.** What are the factors causing collective migration in SoMo? The fact that communities sense and avoid each other over considerable distances means that they react to attractants or repellents that are secreted, or otherwise released, by the parasites. We started by investigating if these factors were produced under normal growth conditions in liquid culture. We concentrated the supernatant of a dense culture (2 × 10⁷ cells ml⁻¹) 10-fold and spotted this "conditioned medium" (CM) next to migrating projections on SoMo plates (Fig. 1a, upper panel). Projections migrating towards the site where CM was spotted reoriented away from it, indicating that it was perceived as a repellent. In a second experiment, CM was either sterile filtered or dialysed (with a 12 kDa cut-off) prior to concentration and spotted next to migrating projections. As seen with untreated CM, filtered CM still repelled migrating projections (Fig. 1a, lower left panel), but dialysed CM lost its repellent properties (Fig. 1a, lower right panel). On the contrary, migrating

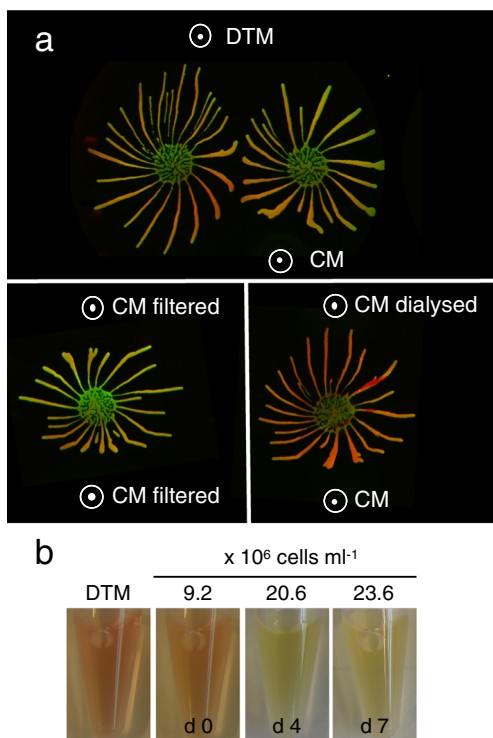

**Fig. 1 Response of migrating cells to fresh medium (DTM) and conditioned medium (CM) after different treatments. a** Upper panel: communities exposed to DTM and CM; lower left panel: communities exposed to filtered CM on both sides; lower right panel: communities exposed to dialysed CM (12 kDa cut-off) or undialysed CM. White circles with dots indicate where solutions were pipetted onto the plates. Community lifts stained for EP (green) and GPEET (red) are shown. **b** Continuous growth in liquid culture acidifies medium. Representative pictures of culture supernatants from early procyclic forms (EATRO1125) grown in DTM for 7 days without dilution. Left, DTM only. Samples were taken on days 0, 4, and 7. The colour change from red to yellow indicates that the culture becomes more acidic with time. Cell titres are indicated.

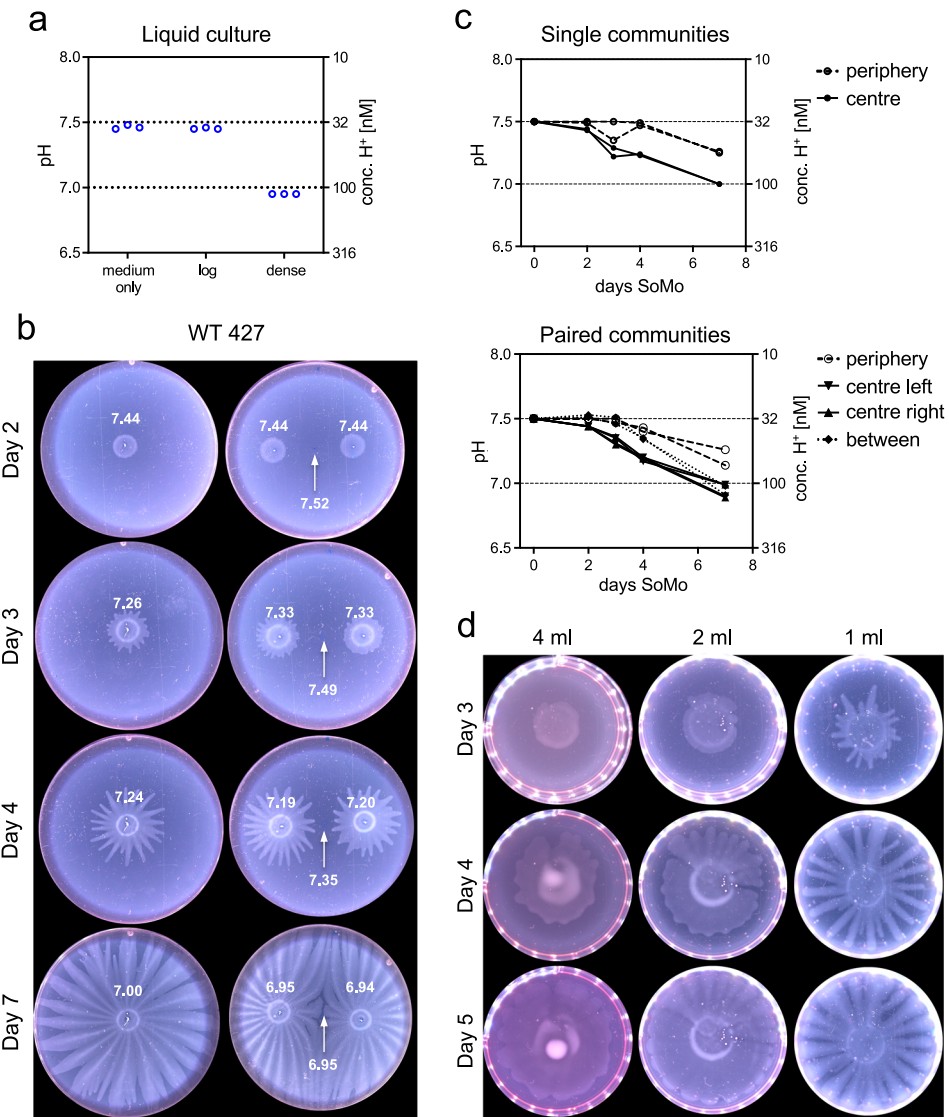

**Fig. 2 Trypanosomes acidify their environment in liquid culture and on semi-solid agarose plates.** Early procyclic forms (Lister 427 Bern) grown in SDM79 were analysed. **a** pH (left $y$-axis) of medium only, a log-phase culture (log, $3.9 \times 10^6$ cells ml$^{-1}$) and a dense culture (dense, $5.6 \times 10^7$ cells ml$^{-1}$). Three technical replicates are shown. The H$^+$ concentration is indicated on the right $y$-axis. Medium only was incubated for the same length of time as the cultures. **b, c** pH measurements on SoMo plates. **b** Trypanosomes ($2 \times 10^5$ cells) were either spotted onto the centre of a plate as a single inoculum (left row) or in pairs (right row). White numbers indicate pH measurements on days 0, 2, 3, 4, and 7 in the centre of a community, at the periphery and in the space between two communities (indicated with a white arrow). **c** Graphs for pH measurement shown in (**b**). Each data point represents the pH measured from individual plates ($n = 2$). Top: plates with single communities. pH at the plate periphery (filled circle, dashed line) or in the centre of the community (filled circle, solid line). Bottom: paired communities. pH at the plate periphery (open circle, dashed line), in the centres of the two communities (filled triangle, solid line), or between them (filled rhombus, dotted line). $Y$-axes are the same as in (**a**). **d** The time-point of migration depends on the volume of the substrate. Equal numbers of early procyclic forms were inoculated onto wells that contained different volumes of medium with agarose. Plates were photographed on days 3, 4, and 5 post-inoculation. Source data are provided as a Source Data file.

parasites were attracted to dialysed CM in the same way that they were to concentrated fresh medium (DTM) (Fig. 1a). These results suggested that repellents might be metabolites that were lost upon dialysis.

**Trypanosomes acidify their environment in liquid culture and on agarose plates.** Trypanosomes secrete a variety of molecules that influence the pH of their environment[31,32]. Indicators for pH, such as phenol red, are commonly used for culture media and it is apparent that dense cultures are more acidic than fresh medium or log phase cultures, as illustrated for early procyclic forms growing in liquid culture (Fig. 1b). We therefore used a small tip micro glass pH electrode to measure the pH in liquid culture and on agarose plates (Fig. 2). Fresh medium has a pH of 7.46 ($\pm$ 0.02) and the supernatant of a log phase culture has a pH of 7.45 ($\pm$ 0.01) (Fig. 2a). The pH decreased to 6.95 ($\pm$ 0.00) when parasites were allowed to overgrow; this corresponds to ~3-fold increase in the concentration of H$^+$ ions (Fig. 2a).

To measure the pH on plates, $2 \times 10^5$ parasites were inoculated either in the centre of a plate (Fig. 2b, left row), or in pairs at a defined distance from each other (Fig. 2b, right row), and the pH was measured after 2, 3, 4, and 7 days post-inoculation. We measured the pH in the centre of a community, at the periphery of a plate and between two communities. The pH at the centres of the communities decreased from 7.50 ($\pm$ 0.01) to 7.31 ($\pm$ 0.05) during the first 3 days and decreased further to 6.96 ($\pm$ 0.05) between days

## a

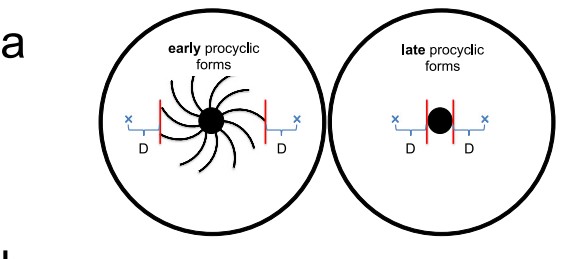

## b

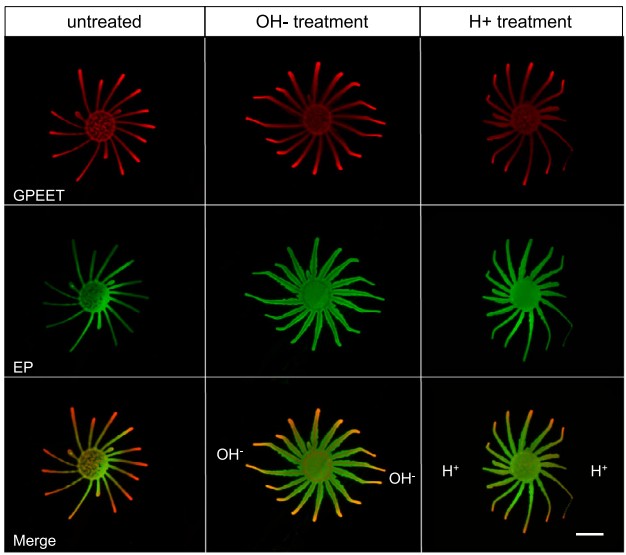

**Fig. 3 Response of early and late procyclic forms to acidic or basic solutions. a** Schematic representation of the pH taxis assay. "x" represents the site where a solution was spotted. "D" is the distance between the closest projection and the spot. In these experiments $D = 1.3$ cm. **b** Early procyclic forms (EATRO1125) on DTM plates were exposed to NaOH (OH− treatment) or HCl (H+ treatment) on both sides of the population. **c** Late procyclic forms were exposed to NaOH on the right side and HCl on the left side of the community. Scale bar: 1 cm. For (**b**) and (**c**) community lifts stained for EP (green) and GPEET (red) are shown. The response of early and late procyclic forms to acidic or basic solutions was tested at least three times.

but rather the need to establish a discernible pH gradient. When the same number of parasites were inoculated onto plates containing different volumes of semi-solid medium, the time point at which projections started to form was inversely proportional to the volume (Fig. 2d). This would be consistent with larger volumes buffering the pH more effectively and requiring more cells, and thus more time, before a gradient was established. We cannot exclude, however, that other factors are also affected by the volume of the medium.

**Reaction of early and late procyclic forms to exposure to acid and alkali.** As a next step, we tested if manipulating the pH gradient had an influence on migration by exposing early procyclic forms to external sources of acid or alkali (Fig. 3a, b). Early procyclic forms were inoculated onto plates and incubated until the projections reached a length of about 2 cm. At this point, solutions of HCl or NaOH were pipetted onto the surface at a defined distance (Fig. 3a) and the plates were incubated for 16–20 h. The migrating cells responded very strongly to both solutions (Fig. 3b, Supplementary Movies 2 and 3). Exogenous NaOH acted as an attractant, causing projections to reorient towards it (OH− treatment), while HCl caused the projections to stop moving outwards, or to move away from the acid (H+ treatment). Using the same assay, another 22 chemicals were tested (Supplementary Fig. 1). In most cases, when a solution was acidic, the projections were repelled, and when a solution was basic, the cells were attracted to it. These experiments also showed that the projections reacted to the pH rather than to specific cations or anions in the solutions.

We next tested late procyclic forms since these do not perform SoMo. Late procyclic forms did not respond to HCl (Fig. 3c, Supplementary Movie 4), but they formed a limited number of projections that migrated towards NaOH. Community lifts (Fig. 3c) confirmed that the parasites were still GPEET-negative and had not dedifferentiated to early forms. In summary, these results demonstrate that procyclic forms exhibit pH taxis, with alkali acting as an attractant for both early and late procyclic forms. In contrast, there are stage-specific differences in their reaction to acid.

**RNA-seq analyses suggest that SoMo is a response to self-generated gradients.** To better understand the mechanisms behind self-organisation of early and late procyclic forms, and their pH responses, we performed RNA-seq under different conditions. A culture of procyclic forms (>60% GPEET-positive) was plated and allowed to form projections (Fig. 4a, far right). A community lift with antibodies against procyclins showed that the tips were strongly positive for GPEET, while the centre of the community stained more strongly for EP. Cells were isolated from the tips and roots of projections (Fig. 4a, far right) and analysed by RNA-seq. Data confirmed that markers of early

## c

3 and 7 (Fig. 2b, c). The pH at the periphery also started to decrease between day 3 and 7, possibly because of diffusion of H+ ions from the parasite communities. When communities were plated in pairs, the pH at the midpoint between them also decreased over time (Fig. 2b, right row, arrow). Thus, communities acidify their vicinity resulting in pH gradients on the plates.

We showed previously that trypanosomes needed to reach a threshold number before they started migrating[18]. We now demonstrate that it is not the cell number per se that is critical,

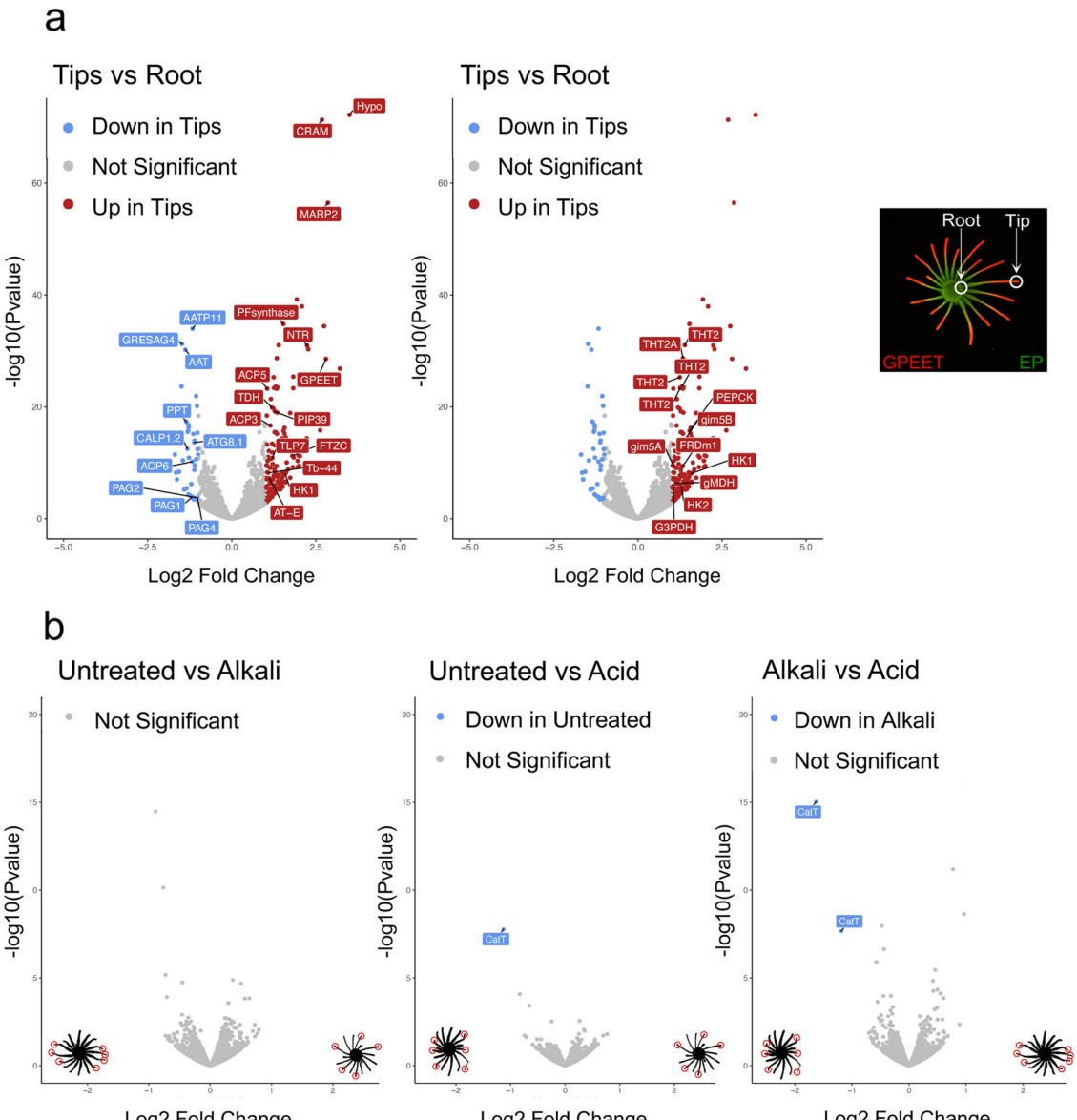

**Fig. 4 Volcano plots comparing transcriptomes of cells from different regions of a community or following exposure to acid or alkali.** DTM plates were inoculated with cells from liquid cultures of early procyclic forms (EATRO1125) in DTM medium. **a** Comparison of the roots and tips of projections. For clarity the same volcano plot is displayed twice with different transcripts annotated. Left: early procyclic form markers, such as GPEET, calflagin, hexokinase 1 and prostaglandin F synthetase are upregulated in tips. Adenylate cyclases such as ACP3, ACP5, and ACP6 are differentially expressed between tip and root. Right: glycolytic enzymes and genes involved in glucose metabolism are upregulated in tips. Far-right panel: community lift of a representative SoMo plate 7 days after inoculation stained for EP (green) and GPEET (red). Tips were isolated when projections were ≥2.5 cm in length. RNA-seq was performed in duplicate. **b** Pairwise comparisons of untreated communities and communities exposed to acid or alkali. Tips were isolated from projections showing a clear response to the treatment (indicated with red circles). RNA-seq was performed in triplicate. The illustrations in the left and right bottom corners of the volcano plots show SoMo communities (in black) and the samples that were collected (red circles). Analysis was done using DESeq2 bioconductor package to identify the differentially expressed genes. The package integrates Wald statistics to identify significantly regulated genes and adjustments were made for multiple comparisons with FDR/Benjamini-Hochberg[69]. Source data are provided as a Source Data file.

procyclic forms were more highly expressed in the tips than the roots (Fig. 4a, left; Supplementary Data 1a). Other transcripts that were more highly expressed in the tip were not characteristic of early procyclic forms in liquid culture, such as cysteine-rich acidic integral membrane protein (CRAM)[33], a microtubule-associated repetitive protein (MARP2)[34] and a hypothetical protein (hypo; Tb927.9.10400). In addition, several adenylate cyclases were differentially expressed on plates; ACP3 and ACP5 were ≥2-fold higher in the tip, while ACP6 and another adenylate cyclase (GRESAG 4; Tb927.11.1480) were ≥2-fold higher in the root. The

up-regulation of high-affinity glucose transporters (THT2 and 2A), glycolytic enzymes and components of the glycosome (Fig. 4a, middle and Supplementary Data 1a) is suggestive of higher glucose metabolism by the tip than the root. Increased expression of these transcripts is also characteristic of early procyclic forms in liquid culture[17,18].

RNA-seq was also performed on the tips of communities exposed to acid or alkali (Fig. 4b and Supplementary Data 1b). There were no significant differences (≥2-fold, $p = 0.05$) between non-exposed communities and those exposed to alkali. A single

cation transporter (CatT, Tb927.11.9000) was more highly expressed in cells exposed to acid, but the read counts were extremely low in both cases. CatT and a copy of a contig (Tb11.v5.0514) were the only significant differences in a comparison of acid- versus alkali-treated communities. Taken together, these results suggest that self-generated pH gradients might be the normal driver of migration on SoMo plates.

**Glucose contributes to the formation of pH gradients.** Since the RNA-seq data indicated that cells at the tip might metabolise glucose more actively than cells at the root, we investigated the role of glucose in SoMo. For these experiments SDM80 medium was prepared with 1 mM glucose, 6 mM glucose, or 6mM N-acetylglucosamine (GlcNAc), which inhibits glucose uptake by binding to hexose transporters[35]. Early procyclic forms of Lister 427 Bern were inoculated onto plates and incubated for 6 days (Fig. 5a). The parasites in SDM80 with 1 mM glucose started migrating later than cells in SDM80 containing 6 mM glucose, and the SoMo defect was even more pronounced when glucose uptake was blocked by GlcNAc. Under these conditions the cells only formed rudimentary "knobs" at the edge of the community and failed to form projections, even at later time points.

The reduced ability to migrate on plates can have a variety of causes, especially in the context of glucose restriction. It might be due to longer population doubling times, motility defects or inefficient energy metabolism. Procyclic cultures of Lister 427 Bern are stably GPEET-positive[27,36]. Nevertheless, we assessed the proportion of early procyclic forms by flow cytometry (Fig. 5b). Regardless of the medium in which the cells were grown, all cultures were ≥97% GPEET-positive. Furthermore, the glucose concentration had no impact on the population doubling time (Fig. 5c) or intracellular ATP levels (Fig. 5d), both assessed in liquid culture. Previous work showed that directional motility is important for SoMo and migration through the fly[29]. We therefore assessed cell motility (Fig. 5e); no differences were observed between cells grown in the presence or absence of glucose.

To investigate whether different glucose concentrations influence the culture pH, we compared medium alone with liquid cultures of trypanosomes in log phase or high density (Fig. 5f). While the pH of log-phase cultures did not differ from that of medium alone, dense cultures of parasites in SDM79 or SDM80 with 6 mM glucose strongly acidified the medium. The concentration of $H^+$ in dense cultures, compared to log phase cultures, increased 2.53-fold (±0.03) for SDM79 and 2.51-fold (±0.06) for SDM80 plus glucose. By contrast, cells grown in SDM80 plus GlcNAc showed a smaller change in $H^+$ ion concentration (1.22-fold ±0.03). We next measured pH changes for communities growing on plates (Fig. 5g). The ΔpH between the centre and the periphery reached −0.31(± 0.02) for parasites on SDM79 and −0.37(± 0.02) for SDM80 + 6 mM glucose, corresponding to a ≥ 2-fold difference in $H^+$ ions at the two locations, whereas plates containing SDM80 + GlcNAc did not show a difference in pH. Taken together, these data show that glucose is required for parasites to generate a pH gradient, and that this correlates with their ability to perform SoMo.

**Cyclic AMP signalling is involved in the response to pH.** RNA-seq identified a number of differentially expressed transcripts, including that of the flagellar pocket protein CRAM, which was more highly expressed in the tips than in the roots (Fig. 4a). CRAM is not essential in procyclic culture forms[37]. When we generated knockouts, these grew at the same rate as their parental line, but two out of three independent clones produced smaller projections. Nevertheless, all three reacted to acid and alkali in the

same way as wild-type cells (Supplementary Fig. 2) so we did not proceed with additional analyses.

Several adenylate cyclases were also differentially expressed between the tip and the root (Fig. 4a). Before we started analysing these and other members of the cAMP signalling pathway, we tested whether PDEB1, which hydrolyses cAMP, had an influence on the pH response. When the PDEB1 null mutant was exposed to alkali or acid, it showed no pH response (Fig. 6a). Notably, the PDEB1 null mutant differs from both early procyclic forms, which react to acid and alkali, and late procyclic forms, which form projections in response to alkali but do not respond to acid (Supplementary Table 1).

Four cAMP response proteins (CARP1-CARP4) were previously identified in an RNAi screen as mediating resistance to an inhibitor of PDEB1[38], but they have not been studied in the context of SoMo. In our attempts to generate knockout cell lines for CARP 1 and 2, we only obtained clones after single rounds of transfection. We did not pursue these further, however, as the resistance genes were not integrated at the correct loci. We were able to obtain knockouts of CARP3 and 4, however (Supplementary Fig. 3a). CARP4 null mutants did not show any difference to the parental line when they were subjected to a standard SoMo assay (Supplementary Fig. 3b) and were not analysed further. Deletion of CARP3, while not affecting growth in liquid culture (Supplementary Fig. 3c) or on plates (Supplementary Fig. 4c, d), or expression of the early marker GPEET (Supplementary Fig. 3d) resulted in a major SoMo defect (Fig. 6b, upper panel) reminiscent of the PDEB1 null mutant. The CARP3 knockout did not display a motility defect in liquid culture (Supplementary Fig. 4a, b) and still reacted to alkali, but did not react to acid on plates (Fig. 6b, lower panel and Supplementary Table 1). An HA-tagged addback of CARP3, expressed from a procyclin locus, partially rescued the SoMo phenotype (Fig. 6b, upper panel), although the projections were shorter than in the parental line. Importantly, the addback was now able to sense and react to both acid and alkali (Fig. 6b, lower panel).

We next assessed the requirement for CARP3 in vivo. For these experiments we generated a null mutant in wild-type *T. b. brucei* Lister 427 using a recently described CRISPR/Cas9 transient expression system[39]. In addition, we engineered an addback ectopically expressing an untagged version of CARP3 (Supplementary Fig. 5). Flies were infected with either the wild type, the knockout or the addback and evaluated for the prevalence and intensity of midgut infections after 14 days. The knockout showed a significantly lower midgut infection rate ($p < 0.00001$, Fisher's exact test, two-sided) when compared to the wild type. In addition, infection intensities were reduced (Fig. 6c). Both the prevalence and intensities of infection with the addback were comparable to the wild type. Thus, the defect in establishing a midgut infection can be attributed specifically to the loss of CARP3. In addition, whereas 55–60% of flies heavily infected with the wild type, and 100% heavily infected with the addback, had trypanosomes in the proventriculus, the knockout gave no heavy midgut infections nor did it colonise the proventriculus (Supplementary Table 2).

CARP3 is present in both the cytoplasm and the flagellum (Supplementary Fig. 4e; www.tryptag.org). Since no interaction partners have been reported, a pulldown was performed with HA-tagged CARP3 (Fig. 6d) and analysed by mass spectrometry. This led to the identification of adenylate cyclases ACP3 and ACP5 in two biological replicates (Supplementary Data 2). These interactions were confirmed by in situ tagging and co-immunoprecipitation of myc-tagged versions of ACP3 and ACP5 in the CARP3-HA addback line (Fig. 6d and Supplementary Data 2). The reciprocal co-immunoprecipitations with anti-myc antibody confirmed the interaction for both ACP3 and

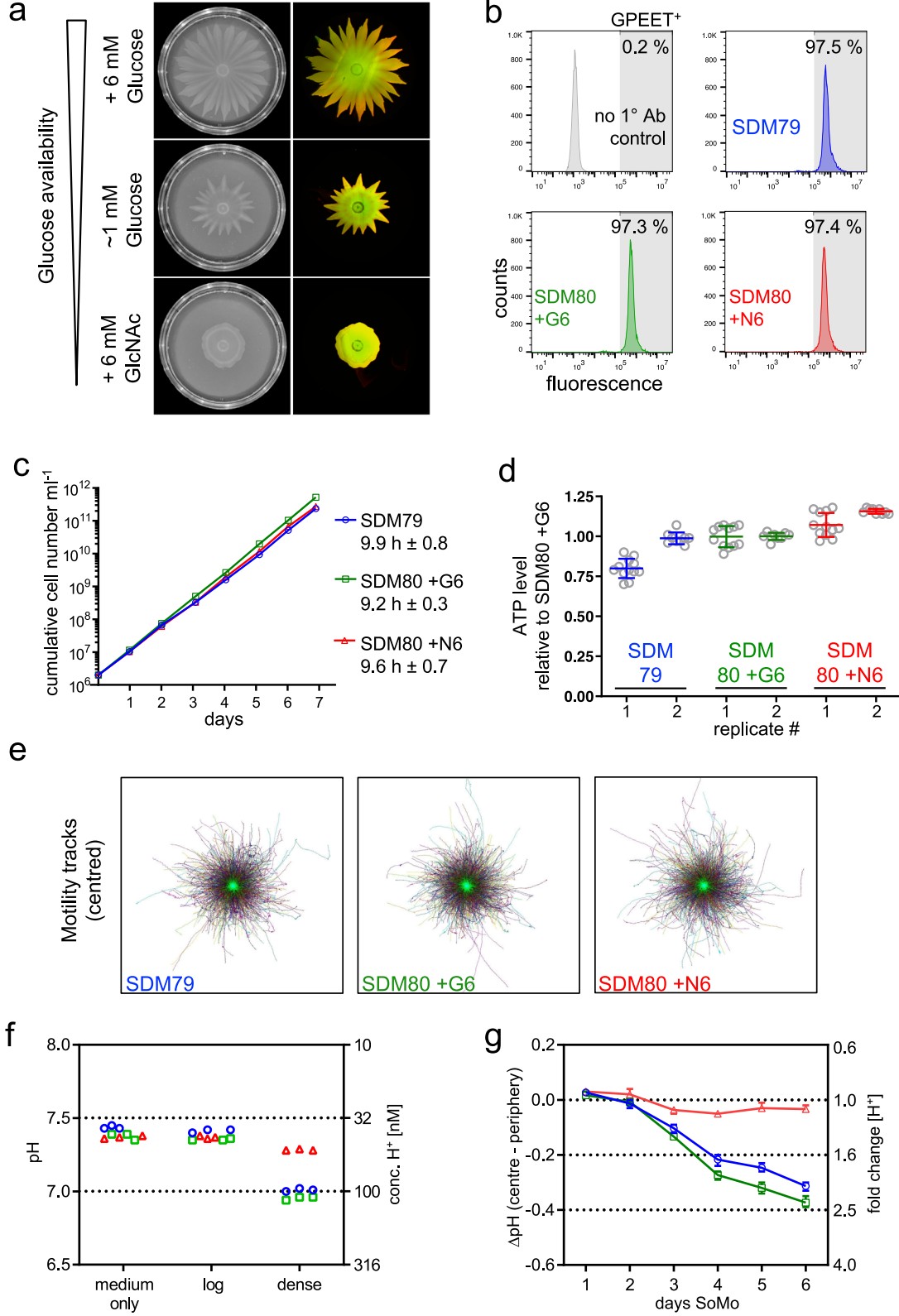

ACP5 (Fig. 6d and Supplementary Data 2). These results, combined with their differential expression in the tips and roots of projections (Fig. 4a) stimulated us to revisit some of the flagellar ACP previously characterised by Lopez and coworkers[26]. In their study they showed that knockdown of ACP1/2 or ACP6 by RNAi resulted in a hyper-SoMo phenotype. We obtained null mutants of ACP2 and ACP6, but it was only possible to generate single knockouts of ACP1 and ACP5 (Supplementary Fig. 6a, 7a) implying that they are essential at this stage of the life cycle. Although we attempted to knock out ACP3, and isolated resistant cell lines, the antibiotic resistance genes were not integrated at the correct locus. The full knockout of ACP6 and the semi-knockout of ACP1 both exhibited hyper-SoMo phenotypes, while the full knockout of ACP2 showed no phenotype (Supplementary Fig. 6b).

**Fig. 5 Glucose availability impacts SoMo by acidification of the culture medium.** Procyclic forms (Lister 427 Bern) grown in SDM79 or SDM80 plus supplements were analysed: SDM79 (blue/open circle), SDM80 + G6 (+6 mM glucose, green/open square), SDM80+N6 (+6 mM GlcNAc, red/open triangle). If not specified, cells were adapted from SDM79 to the medium indicated for one to three weeks. **a** Social motility on semi-solid plates was assessed in SDM80 with varying glucose availability. Left: SoMo plates on day 6 after inoculation, prior to performing community lifts. Cells were adapted from SDM80+G6 to the corresponding liquid medium for 5 days before inoculation. Right: merge of the corresponding community lifts probed for GPEET (red) and EP (green). **b** Flow cytometric analysis. The percentage of GPEET-positive cells (shaded area) is indicated. Cells grown in SDM79 were used as a control (2° antibody only, grey). **c** Cumulative growth in different media. Growth was monitored after transferring cells from SDM79 to the corresponding medium 3 days earlier. Numbers indicate population doubling time averaged over 7 days (mean ± SD). **d** Steady-state ATP levels in different media. Cell lysates were harvested in two independent experiments (replicate 1 or 2, respectively). Cell densities at harvest were between 6.2–8.0 or 7.3–8.9 ×$10^6$ cells ml$^{-1}$, respectively. Appropriate dilutions of cell lysates were assayed; the graph depicts means and error bars (SD) of technical replicates shown as circles ($n = 12$ or 9, respectively). Data are shown relative to SDM80+G6. **e** Directional motility in different media. Cell densities at harvest were between 5.9–6.5 × $10^6$ cells ml$^{-1}$. Centred motility tracks of individual cells are shown at identical scale. >6000 tracks in each medium were pooled from nine movies acquired from three technical replicates. **f** pH measurement of liquid cultures. Culture medium (medium only) and supernatants from log-phase (log) or dense culture (dense). Cell titres per ml in SDM79, SDM80+G6 or SDM80+N6, respectively: 3.2 × $10^6$, 3.8 × $10^6$ or 3.3 × $10^6$ (log) and 5.1 × $10^7$, 4 × $10^7$ or 3.6 × $10^7$ (dense). Three technical replicates are shown for each condition. The corresponding H$^+$ concentration is indicated on right y-axis. **g** pH measurements on SoMo plates inoculated with cells pre-adapted to the corresponding liquid medium. ΔpH (centre−periphery) refers to pH in centre of the community minus the pH at the plate periphery. Each data point represents the mean ΔpH from replicate plates, error bars depict the range ($n = 3$). The fold change in H$^+$ concentration is indicated on the right y-axis. Source data are provided as a Source Data file.

These results are consistent with the findings with the RNAi lines; they also indicate that the hyper-SoMo phenotype in the previously published ACP1/2 RNAi line[26] was due to depletion of ACP1. Unexpectedly, however, and in contrast to the results with RNAi[26], two out of three single knockouts of ACP5 were unable to form projections on plates and the third produced stunted projections (Fig. 6e, upper panel). In terms of their pH response, ACP5 single knockouts resembled CARP3 double knockouts - they were able to form projections migrating towards an alkaline source, but were refractory to acid (Fig. 6e, lower panel and Supplementary Table 1). Further experiments showed that >80% of these cells were GPEET-positive and that the phenotype of the ACP5 mutants was not due to defects in growth or motility (Supplementary Fig. 7b–g). These experiments also revealed that ACP5-myc was localised exclusively at the flagellar tip (Fig. 6f). In summary, we have extended the catalogue of cAMP signalling components that are involved in regulating SoMo. In addition, the finding that PDEB1, ACP5 and CARP3 play a role in the perception of environmental acidity/alkalinity is consistent with SoMo being a response to self-generated pH gradients.

## Discussion

It was proposed more than a decade ago that the collective migration of trypanosomes on plates could be explained by a combination of migration factors and repellents released by the parasites[19], but the identity of these molecules was unknown. Here we show for the first time that trypanosomes acidify their environment as a consequence of glucose metabolism, generating pH gradients by diffusion. While it is commonly assumed that free glucose levels in tsetse are low between blood meals, and this is often stated in reviews, there are no primary publications on this topic. The fact that transcripts encoding high-affinity glucose transporters and glycolytic enzymes are not only upregulated in SoMo tips (Fig. 4) and in early procyclic forms[17], but also in trypanosomes at the beginning of a fly infection[40], is compatible with procyclic forms utilising carbon sources other than proline in tsetse.

Early and late procyclic forms exhibit self-organising properties. It is the perception of these self-generated pH gradients that governs the outward migration of early procyclic forms towards a more alkaline environment, while late procyclic forms remain at the inoculation site. Local acidification of the environment, and the response to it, can also explain the spacing between projections emanating from the same parasite colony, as well as the

cessation of migration or the reorientation of projections from two separate communities on the same plate. In addition, it explains why the start of migration is not solely dependent on a threshold cell number, as was previously reported[18], but rather on the external pH. Early procyclic forms are also able to perceive exogenous sources of acid and alkali as repellents and attractants, respectively. Late procyclic forms are also attracted to a more basic environment, but unlike early procyclic forms they are not repelled by acid, explaining why they remain at the inoculation site on plates and do not perform SoMo.

Analyses of the transcriptomes of cells spontaneously performing SoMo, or reacting to exogenous alkali or acid, are consistent with them being the same response. Comparison of the transcriptomes of trypanosomes at the tips and roots of projections showed that migrating cells at the tip expressed high levels of markers for early procyclic forms. This mirrors what is seen in liquid culture[17,18] as well as at early time points after flies are infected with stumpy forms[40]. Among the adenylate cyclases differentially expressed between the tip and the root, ACP6 is more highly expressed at the root and is also up-regulated in late procyclic forms. By contrast, ACP3 and ACP5, which are more highly expressed in the tip, are not stage-regulated between early and late procyclic forms in liquid culture[17]. This is also true of a few other transcripts, such as CRAM, and may reflect a response to culture on plates.

cAMP signalling is required for the perception of pH gradients (summarised in Fig. 6g). A PDEB1 knockout was not only defective in SoMo[29], but was also impervious to acid and alkali. In contrast to an RNAi mutant of ACP5 which still performed SoMo normally[26], an ACP5 single knockout did not form projections and was insensitive to exogenous acid, but was still attracted to alkali. In this respect, its behaviour resembles that of late procyclic forms. Down-regulation or deletion of ACP6 ([26] and Supplementary Fig. 6b) or deletion of one copy of ACP1 cause hyper-SoMo, while ACP2 is not required for SoMo or pH taxis. ACP1, ACP5 and ACP6 are localised at the flagellar tip ([41] and Fig. 6f), highlighting its importance as a sensor.

For the first time, we show that there is a physical link between a cyclic AMP response protein, CARP3, and adenylate cyclases ACP3 and ACP5. Moreover, the CARP3 null mutant had the same defects in SoMo and acid sensing as the ACP5 single knockout. Interestingly, there is strong evidence that CARP3 is N-myristoylated[42], which could enable it to associate with the inner leaflet of the plasma membrane. N-myristoylation has been shown to regulate signalling in mammals[43] and it is conceivable

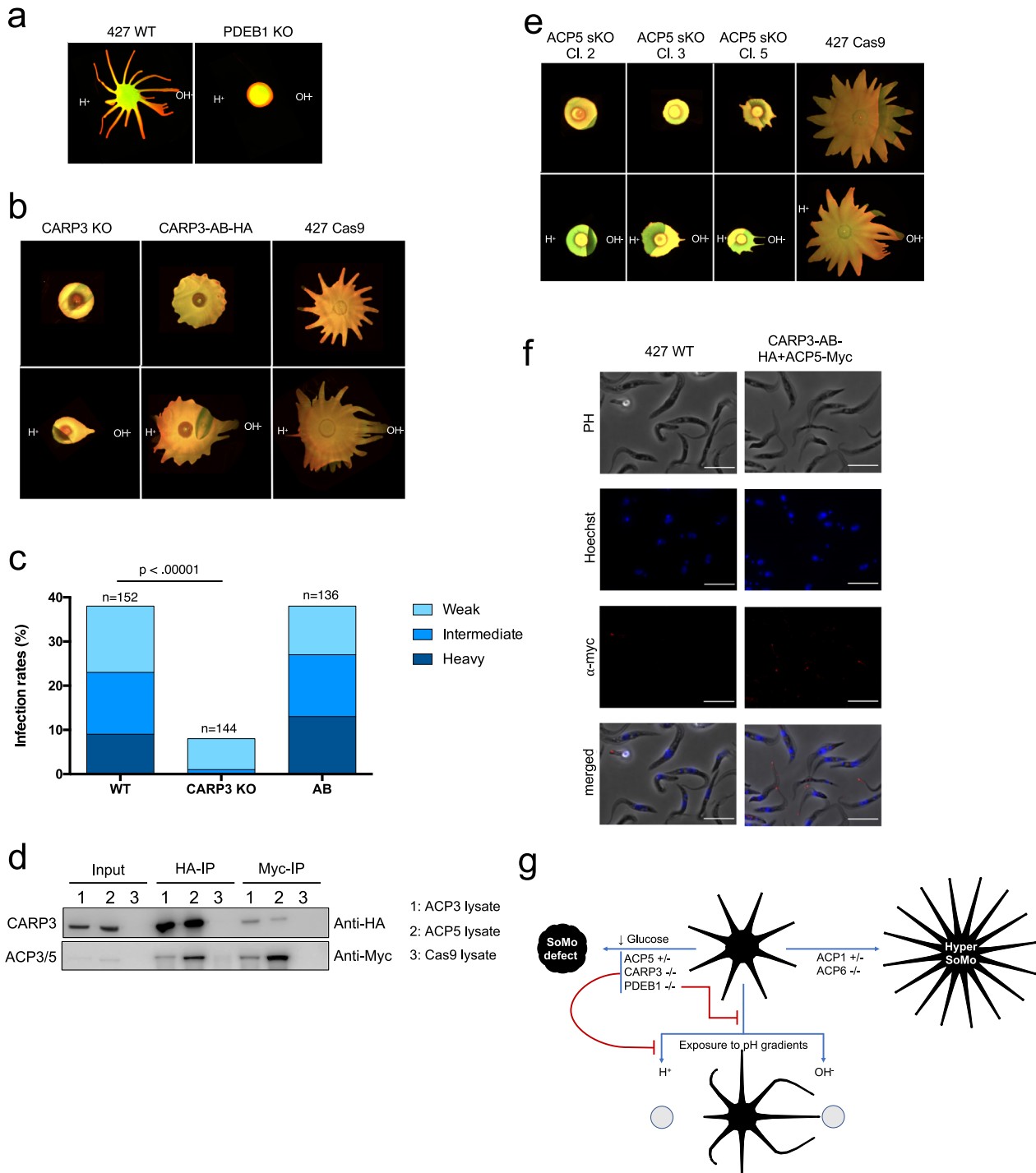

**Fig. 6 CARP3 is required for acid sensing in vitro and for establishment of infection in flies. a** Merged EP and GPEET signals of Lister 427 wt and PDEB1 knockout (KO). **b** Merged EP and GPEET signals of 427 Cas9 parental line, CARP3 knockout and CARP3-HA addback. Upper panel: untreated. Lower panel: exposed to acid and alkali. **c** Teneral flies were infected with wild-type Lister 427 procyclic forms (WT), the CARP3 double knockout (CARP3-KO) and the addback (AB). Dissections were performed 13–14 days post infection. Total numbers are derived from two replicates per cell line (Supplementary Table 2). The intensity of midgut infections was scored as heavy, intermediate or weak as described[36]. The p-value is shown for Fisher's exact test, two-sided ($p$-value = $2.0117^{-10}$). **d** Western blot analysis of immunoprecipitations with anti-HA and anti-myc antibodies (representative blot from three independent experiments). Myc-tagged ACP3 and ACP5 were expressed independently in the CARP3-HA background. The parental line 427 Cas9 was used as a control. Input samples have 40 times fewer cell equivalents than the immunoprecipitated samples. IP: immunoprecipitation with anti-HA or anti-myc antibodies. **e** Merged EP and GPEET signals of the 427 Cas9 parental line and 3 independent ACP5 single allele knockouts; panels as in (**b**). **f** Myc-tagged ACP5 is localised to the flagellar tip in a cell line co-expressing CARP3-HA. Immunofluorescence was performed with an anti-myc antibody (red). DNA was stained with Hoechst dye. Scale bar = 10 µm. **g** Summary of mutants, SoMo phenotypes and responses to pH. −/−: double knockouts; +/−: single knockouts. Source data are provided as a Source Data file.

that acylated and non-acylated forms of CARP3 enable signals to be transmitted from the membrane to the interior of the cell. *T. brucei*, which has the most complex life cycle of all the African trypanosomes, also encodes the largest number of adenylate cyclases. Many of these are stage-regulated, and it is plausible that they have functions in reacting to their local environment. It is also worth noting that CARP3 interacts with other adenylate cyclases, although ACP3 and ACP5 were the top hits (Supplementary Data 2), and that it was recently identified by proximity labelling as interacting with ACP1[44].

pH taxis by trypanosomes is likely to be biologically relevant during fly transmission as there are large differences between the pH of the midgut (pH ~8) and proventriculus (pH ~10.5)[30]. This would enable parasites to follow cues provided by the fly as they migrate to different compartments. The hindgut is also more basic than the posterior midgut[30], and trypanosomes have been detected there, but these have abnormal morphologies[45]. At present we do not know if there are additional chemotactic cues attracting trypanosomes towards to the proventriculus or repelling them from the hindgut. Regardless of whether they migrate in one or both directions, only parasites migrating to the proventriculus can survive and complete the life cycle. We have shown that CARP3 is required for efficient colonisation of tsetse. Interestingly, the knockout has a more severe phenotype than the PDEB1 knockout described previously[29]. The finding that CARP3 interacts with several adenylate cyclases in addition to ACP3 and 5 suggests that it may act as a node for signal perception via cAMP signalling. It is established that procyclic forms respond chemotactically to other effectors[20,21], but their identities are unknown. Chemotaxis may also explain how parasites in the mammalian host disperse from the bloodstream into other tissues[46–48]. Finally, the concept that self-generated gradients help drive directionality and speed expansion into new territories, recently described for bacteria[49], may also apply to trypanosomes.

## Methods

**Trypanosome stocks and culture conditions.** Procyclic forms of EATRO1125 were derived from bloodstream forms of *T. b. brucei* AnTat1.1[50]. Early procyclic forms were cultivated at 27 °C and 2.5% $CO_2$ in DTM supplemented with 15% foetal bovine serum (FBS)[51]. Late procyclic forms were cultured in SDM79[52] containing 10% FBS at 27 °C. Parasites were maintained at a density between $10^6$ and $10^7$ cells ml$^{-1}$. Procyclic forms of *T. b. brucei* Lister 427 and derivatives thereof were cultured in SDM79 or SDM80 containing 10% FBS at 27 °C. SDM80 components were as described[53], except that BME vitamin solution 100x (Bioconcept 5-20K00-H) was used instead of vitamin mix 100x (Invitrogen 010144) and 1% Pen/Strep stock solution (Bioconcept 4-01F00-H) was used instead of 0.1 mM kanamycin. Glucose-free SDM80 medium was supplemented with 10% FBS, which results in a final concentration of ~1 mM glucose. If indicated, SDM80 + 10% FBS was supplemented with 6 mM D-glucose or 6 mM N-Acetyl-D-glucosamine (GlcNAc) from a 1.2 M stock solution in water. Population doubling times were determined over a period of at least 5 days in which the cell density was determined daily.

**SoMo assays and community lifts.** The protocol to produce plates was adapted from Imhof and coworkers[18]. Plates were always used on the same day that they were poured. Culture medium (10 ml) supplemented with 0.4% agarose (Promega V3125) was poured into petri dishes (85 mm diameter). Open plates were then air-dried for 1 h in a laminar flow hood. Cells from an exponentially growing culture were centrifuged briefly and resuspended in the residual medium at a density of 4 × $10^7$ cells ml$^{-1}$. Two hundred thousand cells were spotted per inoculum; plates were sealed with parafilm and incubated in a humidified environment at 27 °C and 2.5% $CO_2$. Photographs of plates were taken using a Nikon E8400 camera.

Community lifts for the detection of GPEET and EP procyclins were performed as described[18], using nitrocellulose filters (Whatman Protran BA85), K1 rabbit anti-GPEET[36] at a dilution of 1:1000 and TBRP1/247 mouse anti-EP (Cat. no. CLP001A, Cedarlane Laboratories, Burlington, Canada) at a dilution of 1:2500 as primary antibodies (1 h incubation time at room temperature). The secondary antibodies goat anti-mouse IRDye 800CW (LI-COR Biosciences, Bad Homburg, Germany) and goat anti-rabbit IRDye 680LT (LI-COR Biosciences) were used at dilutions of 1:10000 (1 h incubation time at room temperature). The filters were rinsed 3 times in TBS + 0.05% Tween (Sigma), followed by one final rinse in PBS before being dried overnight at room temperature and scanned on a LI-COR

Odyssey Infrared Imager model 9120, using Odyssey Application Software, Version 3.0.21.

**pH measurements on SoMo plates and in liquid cultures.** To monitor pH changes in liquid culture and on agarose plates, trypanosomes were cultured in a humidified incubator at 27 °C and 2.5% $CO_2$. SoMo plates were kept in the same incubator, without sealing the lids with parafilm. For analysis of growth rate and pH measurements, cells were maintained between 1–10 × $10^6$ cells ml$^{-1}$ by daily monitoring and dilution. "Log-phase" samples had densities of 3–9 × $10^6$ cells ml$^{-1}$. If not stated otherwise, "dense" samples were obtained from cultures with a concentration of 1–1.5 × $10^7$ cells ml$^{-1}$ without further dilution for 2 days. "Medium only" refers to complete medium incubated in a culture flask in a humidified incubator at 27 °C and 2.5% $CO_2$ for at least 12 h. The pH of culture supernatants or SoMo plates was measured using a microelectrode (Thermo Scientific™ Orion™ 9810BN Micro pH Electrode; fisher scientific).

**Preparation of concentrated supernatant.** Early procyclic forms were grown until they reached a concentration of 3 × $10^7$ cells ml$^{-1}$. The cells were centrifuged for 5 min at 2500 *g* and the conditioned medium was transferred to a new tube. Conditioned medium was either concentrated directly or first sterile filtered (Jet Biofil syringe filter, 0.22 μm, REF: FVP203030K) or dialysed. 2 ml of conditioned medium were dialysed with dialysis tubing cellulose membranes (Sigma-Aldrich, cut-off 12 kDa, REF: D9777-100FT) against 50 ml DTM supplemented with 15% FBS at 4 °C in two rounds (16 h, then 12 h). To concentrate the conditioned medium, 1 ml was transferred into a 1.5 ml Eppendorf tube and vacuum centrifuged for several hours until 100 μl remained in the tube. As a control, DTM only was concentrated in parallel.

**pH taxis assay.** Procyclic forms were inoculated onto plates as described for SoMo assays. Following incubation at 27 °C for four days, concentrates of 30 μl conditioned medium or medium only, or different chemical solutions were added to the plates at a distance of 1.5 cm from migrating projections (Fig. 3a, D = distance). The plates were sealed with parafilm and incubated at 27 °C for 16–20 h. Stock solutions were prepared at a concentration of 1 M, if not stated otherwise.

**Hot phenol extraction of RNA from migrating tips.** Plates containing semi-solid DTM medium were inoculated with 2 × $10^5$ cells from a culture of procyclic form EATRO1125 (60–80% GPEET-positive). Following incubation for 6–7 days, tip and root samples were collected from projections that were ≥2.5 cm long. For collecting tips reacting to acid or alkali, 3 × $10^5$ cells were seeded, incubated for 5 days and then 30 μl 1 M NaOH or 1 M HCl were added on two sides of migrating cells at a distance of 1.5 cm from the tips (Fig. 3a, D = distance). In total, between 70–100 tips of migrating projections, that were exposed to acidic or alkaline solutions for 16 h, were harvested. Only tips that clearly showed a reaction were collected (usually 5–7 tips per community).

Cells were harvested by punching out a piece of agarose using an inverted P2 micro tip (outer and inner diameters 6 and 4.5 mm, respectively). The tip subpopulation refers to the outermost 3 mm of protrusions. The root refers to the region at the base of the projection as shown in Fig. 4a. Between 20–25 tips or 3–5 root samples were transferred into 1 ml DTM in a 24-well plate well. Cells were washed off the agarose pieces by careful rinsing with a P1000 pipette and the cell suspension was transferred to a 1.5 ml Eppendorf tube. The agarose pieces were rinsed once more with 0.5 ml DTM, added to the cell suspension and centrifuged for 2 min at 3300 × *g*. On average, approximately 1 × $10^5$ cells were retrieved from each tip and 6 × $10^5$ cells from each root sample. For tips reacting to acid or alkali, the wells of a 6-well plate contained a larger number of samples (up to 100 tips). In this case, the supernatant was carefully aspirated from the well and collected in a 15 ml Falcon tube and the agarose pieces were washed twice with 4 ml medium. The washes were added to the cells in the 15 ml Falcon tube and centrifuged for 5 min at 2500 × *g*. Cell pellets were resuspended in 50 μl residual medium and RNA was extracted with hot phenol as described[54].

**RNA-seq analysis.** Total RNA (maximum 10 μg) was processed and used for Illumina sequencing as described previously[17]. Poly(A)-selected RNAs were used for the preparation of cDNA libraries using an Illumina TruSeq kit. Sequencing of cDNA libraries was performed at Fasteris, Geneva, using Illumina HiSeq sequencing systems with 150 bp read lengths and sequencing depths of approximately 20 million reads per sample. Reads were mapped to the *T. b. brucei* 927 reference genome version 5.1, using the bowtie2 tool available in the Galaxy Interface (usegalaxy.org) with default parameters that allow a maximum of 2 mismatches per 28 bp seed (Galaxy version 1.1.2). Read counts were extracted using "featureCounts" tool available in Galaxy using TritrypDB-41 GFF annotation file. DESeq2 analysis were performed to identify the differentially regulated genes and RPM values were calculated to estimate the transcripts abundance. Raw read files are deposited at the European Nucleotide Archives (ENA) http://www.ebi.ac.uk/ena as study PRJEB41935.

**Flow cytometry**. Cells were fixed in 2% (w/v) PFA/PBS at 4 °C overnight at a concentration of $1 \times 10^7$ cells ml$^{-1}$ and processed for flow cytometric analysis as previously described[55,56]. Rabbit antiserum α-GPEET K1[36] and Alexa Fluor 488-conjugated secondary antibody (Invitrogen, Thermo Fisher Scientific) were used at 1:1000 dilution in 2% BSA/PBS and incubated for 1.5 h at 4 °C with rotation. To remove particles of subcellular size, a cut-off of $7.5 \times 10^5$ was applied to the forward scatter. A total of ten thousand events were recorded using a ACEA NovoCyte flow cytometer and analysed without gating using FlowJo software version 7 & 10 (BD Life Sciences).

**Determination of steady-state ATP levels**. ATP concentrations were measured based on a previously described ATP production assay[57]. To measure steady-state ATP levels in whole cells, cells were washed twice in ice-cold PBS and resuspended in PBS at $2.5 \times 10^7$ cells ml$^{-1}$. Denaturation with perchloric acid, precipitation and neutralisation with KOH was performed as described[57]. The cleared lysate was serially diluted 4-fold in H$_2$O. Routinely, cells were harvested in triplicate and the second dilution of the cleared lysate (16-fold dilution) was used for the bioluminescence assay. For each replicate, ATP concentrations were measured in triplicate. ATP concentrations were determined using the ATP Bioluminescence Assay Kit CLS II (Roche 11699695001, from Sigma-Aldrich) according to manufacturer's instructions. For each dilution, 10 μl was added to 40 μl of 0.5 M Tris-acetate, pH 7.75 and supplemented with luciferase reagent to a final volume of 100 μl. Titrations of the ATP standard supplied were analysed in parallel. Luminescence was measured in 96-well microtitre plates (opaque, non-binding) using a Turner Bio-Systems Modulus Microplate Luminometer, Model 9300-001. Signals were integrated over 5 sec in steady-glo mode. Three readings per plate were acquired, with a period of 10 min. Values of the second reading were used for analysis, since only the second and third readings yielded stable signals.

**Motility assays**. Cultures were allowed to grow to a density between $4–7 \times 10^6$ cells ml$^{-1}$ and adjusted to a final concentration of $5 \times 10^6$ prior to the experiment. These cells (in medium) were placed into motility chambers[58]. To prevent cell adherence, the chamber was precoated with 1% poly L-glutamic acid (Fisher Scientific, P4886-25MG) for 1 h and washed twice with 70% EtOH. The chambers were then washed twice with 100 μl medium and once with cell suspension before loading 100 μl of cells at a concentration of $5 \times 10^6$ cells ml$^{-1}$. The edges of the coverslip were sealed to avoid evaporation and capillary flow of liquid. Time lapse movies were recorded at 200 ms, with 250 cycles and ×10 magnification with a Leica DM 5500B microscope. The dataset was analysed with a tracking macro provided by Richard Wheeler, University of Oxford[59]. Velocity, speed, and individual cell tracks were plotted using Fiji version 1.0[60] and Prism Version 6.

**Immunoprecipitation and proteomic analysis**. Isolation of HA-tagged protein complexes was performed as previously described[61] using the addback cell line expressing CARP3-HA, or derivatives expressing CARP3-HA and either ACP3-myc or ACP5-myc. Myc-tagged proteins were immunoprecipitated with 5 μg monoclonal antibody 9E10 (Cat No: MA1-980) and Dynabeads™ Pan Mouse IgG (Cat No: 11041), both from ThermoFisher Scientific (Invitrogen). Isolated protein complexes were subjected to Western blot analysis, or protein bands were cut from Coomassie-stained polyacrylamide gels, subjected to trypsin digestion and proteins were identified by LC-tandem mass spectrometry[62] at the Proteomics and Mass Spectrometry Core Facility of the University of Bern.

**Generation of knockouts and addbacks**. Single knockouts, double knockouts and tagged addbacks used for all in vitro experiments were generated in the parental 427 Cas9 cell line[29]. The CARP3 double knockout used for fly experiments was generated in T. b. brucei Lister 427[36] using transiently expressed Cas9 and T7 RNA polymerase[39]. An addback ectopically expressing CARP3 was derived from the null mutant. Targeting cassettes were amplified from pPOTv7-hygromycin and pPOTv7-G418[63] plasmids. The primers used for PCR amplification of the targeting fragments and the sgRNA templates were designed using LeishGEdit. These primers and the primers used for genotyping PCRs can be found in Supplementary Data 3. The knockouts and the addback were verified by PCR (Supplementary Data 3). To generate addbacks, HA-tagged and untagged versions of CARP3 were cloned into the pGAPRONE-ΔLII[64] derivate pG-mcs-puro[65], linearised with Spe I and integrated upstream of a procyclin locus on chromosome 6. Transfection was performed as described previously[66] using TbBSF transfection buffer[67]. Stable transformants were selected using 15 μg ml$^{-1}$ Geneticin G418 sulphate, 25 μg ml$^{-1}$ hygromycin and 1 μg ml$^{-1}$ puromycin.

**Immunofluorescence analysis (IFA) of HA- and or Myc-tagged cells**. Cells were harvested at a density of $2–5 \times 10^6$ cells and then washed once with PBS. They were left to settle on a coverslip for 20 min and then fixed with 4% PFA in PBS for 4 min. After removing the supernatant, the cells were permeabilized with Triton-x 100 diluted to 0.2% in PBS for 5 min and the liquid was aspirated. The samples were blocked in PBS + 4% BSA for 1 h at RT. The primary antibodies (monoclonal α-HA rat 3F10 and monoclonal α-myc mouse 9E10, from Roche, distributed by Sigma Aldrich) were diluted 1:1000 and 1:500 respectively in PBS + 0.4% BSA and the samples incubated for 1 h at RT. The cells were then washed 3 times with PBS

and the supernatant aspirated after every washing step. They were then incubated with the secondary antibodies AlexaFluor 488 conjugated donkey α-rat (Life Technologies) and AlexaFluor 488 conjugated goat α-mouse (Invitrogen/Thermo Fisher) both diluted 1:2000 in PBS + 0.4% BSA for 1 h at RT. The samples were then washed 3 times as described for the primary antibodies and incubated with Hoechst 33342 (20 μg/ml) diluted 1:500 in PBS for 2 min. After three washing steps, the cover slips were mounted on microscope slides with Mowiol and imaged with a Leica DM 5500B microscope with a ×100 objective. The acquired images were processed with Fiji[60].

**Fly infections**. Glossina morsitans morsitans pupae were purchased from the Department of Entomology, Slovak Academy of Science, Bratislava, Slovakia or were a kind gift from Jan van den Abbeele, Institute of Tropical Medicine, Antwerp, Belgium. Teneral flies were infected by membrane feeding with $2.5 \times 10^6$ parasites ml$^{-1}$ and maintained on horse blood as described[36], except that pupae and flies were kept at 24 °C, 65–70% humidity, rather than 27 °C in the original protocol. Flies were dissected and midgut infections were analysed on days 13 and 14 post-infection. The intensities of infections were graded as described[36].

**Reporting summary**. Further information on research design is available in the Nature Research Reporting Summary linked to this article.

## Data availability
The RNA-seq data generated in this study have been deposited at the European Nucleotide Archives (ENA) under accession code PRJEB41935. The mass spectrometry proteomics data have been deposited to the ProteomeXchange Consortium via the PRIDE[68] partner repository with the dataset identifier PXD030766. Source data are provided with this paper.

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

## Acknowledgements

Berta Pozzi, Stephanie DeMarco and Boris Striepen are thanked for stimulating discussions and constructive comments on the manuscript. We are indebted to Jan van den Abbeele, Institute of Tropical Medicine, Antwerp for a gift of tsetse pupae. This research was supported by the Swiss National Science Foundation to I.R. (Grant numbers 310030_184669 and 31003A_166427) and the Canton of Bern.

## Author contributions

S.S., S.K., A.N., D.A. and I.R. conceived, designed and conducted experiments. R.E. and M.B. conducted experiments. S.S., S.K., A.N., D.A., M.B., R.E. and I.R. analysed the data. S.S. and I.R. wrote the paper.

## Competing interests

The authors declare no competing interests.
