## [Peer Review File · Nature Communications]

Cyclic AMP signalling and glucose metabolism mediate pH taxis by African trypanosomesEditorial Note: This manuscript has been previously reviewed at another journal that is not operating a transparent peer review scheme. This document only contains reviewer comments and rebuttal letters for versions considered at Nature Communications.

Reviewers' Comments:

Reviewer #4:

Remarks to the Author:

The authors have addressed several of the comments raised in the previous round of review, with changes in the text and addition of some additional data. Some questions remain only partially answered and deemed reasonably feasible for the research group to be addressed.

Understanding the argument about current challenges of individual cell motility measurements, the SoMo assays would benefit a more systematic inclusion of all lines (ACP5+/-, CARP3-/- and PDEB1-/-) for measurement of taxis (as in supplemental table 2 for PDEB1-/-) and addressing impact of the knockout on growth on plates (as in supplemental figure 4c for CARP3-/-).

While taxis studies in the fly may indeed be challenging, in vitro plate motility assays in alkaline conditions that mimic the physiological pH gradients in the fly should be more straightforward and relevant to the defective phenotype in the fly. Can the CARP3 mutant phenotype be rescued with reduced glutathione? If the knockout is hampering taxis in the fly glut, adding reduced glutathione to the infectious blood meal would be anticipated not to impact on the phenotype.

Even though not a full knockout, the ACP5+/- strain displays similar characteristics in the SoMo plate/taxis assay which would further support the conclusions, and therefore this should be rather straightforward to be tested and included.

Reviewer #4 (Remarks to the Author):

The authors have addressed several of the comments raised in the previous round of review, with changes in the text and addition of some additional data. Some questions remain only partially answered and deemed reasonably feasible for the research group to be addressed.

Understanding the argument about current challenges of individual cell motility measurements, the SoMo assays would benefit a more systematic inclusion of all lines (ACP5+/-, CARP3-/- and PDEB1-/-) for measurement of taxis (as in supplemental table 2 for PDEB1-/-) and addressing impact of the knockout on growth on plates (as in supplemental figure 4c for CARP3-/-).

In our previous revision we provided data for the two mutants that were most relevant to answer the questions posed by the referee. In terms of the ACP5+/- and CARP-/- it is possible to see the

contours of the colonies stay curved in the presence of acid, where the wild-type and addback retract or flatten (Fig 6b and 6e). Measurements for these mutants are now provided, as are growth on plates for ACP5+/- and PDEB1-/- (Table S2 and Figure S7f and 7g).

While taxis studies in the fly may indeed be challenging, in vitro plate motility assays in alkaline conditions that mimic the physiological pH gradients in the fly should be more straightforward and relevant to the defective phenotype in the fly.

Although it would indeed be interesting to see how individual cells behave in response to pH gradients, these assays are not straightforward at all. In our previous rebuttal we indicated that it was not easy to do and cited the work of the Engstler lab. To be more explicit, a first publication on tracking single cells in SoMo appeared in August this year. It requires fluorescently labeled cells (approximately 5% in a background of unlabeled cells). Not only would this require making derivatives of each mutant, and setting up appropriate conditions, the technology is not so advanced that it can provide meaningful results just yet.

To cite from the abstract of this publication (Krüger et al., PMID: 33755816 Single-cell motile behaviour of *Trypanosoma brucei* in thin-layered fluid collectives):

“Therefore, for the first time, we analyse the motility behaviour of trypanosomes directly in a widely used assay, which aims to evaluate the parasites behaviour in collectives, in response to as yet unknown parameters. In a step towards understanding whether, or what type of, swarming behaviour of trypanosomes exists, we customised the assay for quantitative tracking analysis of motile behaviour on the single-cell level. We show that the migration speed of cell groups does not directly depend on single-cell velocity and that the system remains to be simplified further, before hypotheses about collective motility can be advanced.”

Once our manuscript is published, one of the “as yet unknown parameters” will be known. Quite apart from the fact that my lab will close in January (for the reviewer’s information, due to mandatory retirement in Switzerland), the Engstler lab is unquestionably the best equipped to tackle the downstream experiments, which are not trivial.

Another point that we wish to clarify: we have clearly shown that procyclic forms react to endogenous and exogenous pH gradients. Although pH measurements were made for the midgut and the proventriculus, we do not know the Δ pH between the midgut lumen and the ectoperitrophic space, or how this changes upon infection - as we mentioned in our previous rebuttal, this would be a whole new project. It is also not known what Δ pH the trypanosomes are actually exposed to, since the peritrophic matrix may dampen the steepness of the gradient between the midgut and proventriculus. What we can definitely say is that pH sensing, in the way it is manifested by procyclic forms, is biologically relevant. It would be different if the pH through the alimentary tract were constant, or if the proventriculus was more acidic than the midgut.

Can the CARP3 mutant phenotype be rescued with reduced glutathione? If the knockout is hampering taxis in the fly gut, adding reduced glutathione to the infectious blood meal would be anticipated not to impact on the phenotype.

This is a new experiment suggested by the reviewer in the second round of review. With the best will in the world, we do not understand the rationale for it. Glutathione is an antioxidant that prevents reactive oxygen intermediates from killing trypanosomes and increases the incidence of midgut infections with wild-type trypanosomes (in our hands from about 40% to >95%). We do not use it when we analyse mutants, because we want infections to be as natural as possible. It is

possible that glutathione would increase infection rates with the CARP3 knockout for the same reason as for the wild type, by reducing killing in some flies. Therefore, neither outcome (unchanged or increased prevalence) would add to or detract from the data we provide.

Even though not a full knockout, the ACP5^{+/-} strain displays similar characteristics in the SoMo plate/taxis assay which would further support the conclusions, and therefore this should be rather straightforward to be tested and included.

The Editor agrees that this is beyond the scope of this paper.